# Recent Advances in nccRCC Classification and Therapeutic Approaches

**DOI:** 10.3390/cells14221781

**Published:** 2025-11-13

**Authors:** Hewei Wang, Yiyuan Chang, Kaiyan Wang, Rong Liu

**Affiliations:** Translational Cancer Research Center, Peking University First Hospital, Beijing 100034, China

**Keywords:** non-clear cell renal cell carcinoma (nccRCC), molecular classification, treatment

## Abstract

**Highlights:**

**What are the main contents summarized?**

**Abstract:**

Non-clear cell renal cell carcinoma (nccRCC) constitutes a biologically diverse category of renal malignancies. The 2022 WHO classification framework has significantly evolved to incorporate molecularly defined entities alongside traditional histologic subtypes, reflecting the growing recognition of distinct pathogenic drivers. Current therapeutic paradigms for advanced disease remain suboptimal, with treatment strategies often extrapolated from clear cell renal cell carcinoma (ccRCC). In this review, we highlight transformative multi-omics approaches to address nccRCC’s profound heterogeneity, which enables molecular stratification beyond conventional pathology, identifying novel subtypes characterized by unique immune microenvironment features, metabolic profiles, and genomic instability patterns. This molecular reclassification provides a foundational framework for precision oncology, facilitating patient selection for targeted therapies and immunomodulatory strategies. Advancements in multi-omics subtyping represent a pivotal shift toward biologically guided clinical management and underscore the imperative for biomarker-driven therapeutic development in nccRCC.

## 1. Introduction

Renal cell carcinoma (RCC) ranks as the fourteenth most common cancer globally and it includes multiple heterogeneous subtypes, the majority of which originate from renal tubular epithelium [1,2]. In 2022, an estimated 434,419 new cases of RCC were diagnosed globally, accounting for 2.2% of all cancer cases. RCC was responsible for approximately 155,702 deaths worldwide, representing 1.6% of all cancer-related fatalities. Men have a higher incidence than women, and the risk increases with age. Although RCC mortality has been relatively stable over the past decades, recent data indicate a rising incidence, particularly among younger individuals [3]. ccRCC, the most common subtype of RCC, represents about 70–80% of cases. The remaining 20% are collectively categorized as nccRCC, among which papillary renal cell carcinoma (pRCC) is the most common (13–20%), followed by chromophobe renal cell carcinoma (chRCC) (8–12%) [4]. After the 2022 World Health Organization (WHO) classification update, changes occurred in nccRCC classification. The new classification introduced a molecular-based system, highlighting the significance of molecular analysis in diagnosis. This includes several molecularly-defined renal tumor entities, such as SMARCB1-deficient renal medullary carcinoma, TFEB-altered RCC, ALK-rearranged RCC, and ELOC-mutated RCC [5].

Clinically, the treatments of patients with nccRCC, despite significant heterogeneity, still largely follow ccRCC guidelines, yet with much lower efficacy. The classic targeted therapy sunitinib achieves an objective response rate (ORR) of 27% in ccRCC, while only 4% in pRCC [6]. Immunotherapy—the combination of nivolumab and ipilimumab—yields a higher ORR of 42% in ccRCC [7], versus 19.6% in nccRCC [8]. Given the suboptimal outcomes of current therapies in nccRCC, there is an urgent need for molecular stratification based on drug sensitivity to enable personalized treatment approaches for advanced-stage patients.

This article aims to review the latest literature on nccRCC classification and treatment, guiding clinical practice and enhancing understanding of nccRCC.

## 2. Methodology

A systematic literature search was conducted using PubMed and ClinicalTrials.gov for records from 1 January 2000, to 10 July 2025. The initial core search for “Non-clear cell renal cell carcinoma” in titles/abstracts identified 479 records (Full list of MESH terms in Appendix A). A supplementary search for key histological subtypes (“papillary renal cell carcinoma”, “Chromophobe Renal Cell Carcinoma”, “Collecting duct carcinoma”) in titles over the past decade yielded 909 additional publications (Appendix B). A parallel search of ClinicalTrials.gov using “non-clear cell renal cell carcinoma” and “treatment” identified 268 trials. To capture authoritative guidelines and the most recent research findings, we also manually screened the 2022 WHO classification of renal cell tumors, as well as the meeting libraries of ASCO/ESMO.

The study selection process was governed by explicit inclusion and exclusion criteria. We included prospective clinical trials, large retrospective studies, relevant clinical guidelines, and significant conference abstracts. Studies were excluded if they focused primarily on ccRCC or unspecified RCC, were non-English publications, or were very small sample-sized case reports to enhance the overall quality and validity of the evidence synthesized.

The screening process was performed independently by two investigators (Hewei Wang, Yiyuan Chang). After duplicate removal, records were screened by title/abstract and then by full text. Discrepancies were resolved through consensus or a third reviewer. Reference lists of included articles were hand-searched for additional publications. The entire process adhered to PRISMA guidelines, with a flow diagram presented as Figure 1. Data on study characteristics and outcomes—primarily median overall survival, objective response rate, and progression-free survival—were systematically extracted.

## 3. Pathology and Genetics in nccRCC

According to the clinically established classification criteria, nccRCC is classified into several subtypes, including papillary renal cell carcinoma, Chromophobe renal tumors, Collecting duct carcinoma (CDC), renal medullary carcinomas (RMC), and MiT family translocation RCC (tRCC), etc. An overview of the most frequent subtypes, as well as their implications for treatment decision-making, are listed as following (Figure 2).

### 3.1. Papillary Renal Cell Carcinoma

pRCC which originates from the proximal tubular epithelial cells, is the most frequent nccRCC subtype, accounting for approximately 15% of all RCC cases [4]. It was traditionally classified into two subtypes based solely on morphology. Type 1 pRCC is histologically defined by papillae covered with a single layer of columnar epithelial cells [9]. In contrast, type 2 pRCC exhibits more aggressive histological features, including large cells with eosinophilic cytoplasm and pleomorphic nuclei. Clinically, type 1 pRCC is associated with a more indolent course and a better prognosis than both type 2 pRCC and ccRCC [10].

However, since the 2022 WHO fifth edition classification, pRCC is recognized as a complex spectrum of histological and molecular features, rendering the former type 1 and type 2 classifications obsolete [5]. Genetically, type 1 pRCC is frequently driven by MET proto-oncogene mutations or chromosome 7 amplifications, whereas type 2 exhibits greater heterogeneity, involving alterations in FH, TFE3, CDKN2A/2B, and others [11,12]. Notably, a subset of type 2 cases displays a CpG island methylator phenotype, which correlates with poor survival and may have therapeutic implications [13].

### 3.2. Chromophobe Renal Tumors

chRCC originates from the intercalated cells of the collecting duct, accounting for approximately 5% of all RCC cases [14]. Notably, it is the most common nccRCC in young women. Histologically, it is characterized by large cells with pale, flocculent cytoplasm and prominent “plant-like” cell membranes [15].

chRCC is characterized by frequent whole-chromosome losses, notably chromosomes 1, 2, 6, 10, 13, 17, and 21, occurring in about 80% of cases [15]. WES identifies recurrent mutations in tumor suppressor genes TP53 and PTEN, with PTEN alterations significantly linked to poor survival. Additionally, mutations in genes such as TSC1, TSC2, and MTOR implicate mTOR pathway activation [16].

### 3.3. Collecting Duct Carcinoma

CDC, a highly aggressive RCC subtype accounting for less than 2% of RCC, originates from the medullary collecting ducts and has a dismal median survival of approximately 44 weeks [17,18,19]. Microscopically, CDC is characterized by two main histological features: a desmoplastic stroma and a glandular part composed of atypical epithelium [20]. However, its diagnosis can be challenging due to histological overlap with entities such as renal medullary carcinoma (RMC) and FH-deficient RCC [21].

Immunophenotypically, CDC typically expresses PAX8, 34βE12, and SMARCB1 (INI-1), and the tumor cells are typically negative for OCT3/4 [22,23]. Molecularly, loss of heterozygosity is associated with a poor prognosis in CDC. Chromosomal alterations and deletions on 1q, 8p, and 13q have been implicated in CDC pathogenesis [19]. A next-generation sequencing (NGS) study revealed genomic alterations in SETD2, CDKN2A, SMARCB1, and NF2 in CDC [24]. Crucially, CDC is not driven by angiogenesis and is refractory to VEGF-targeted therapies, necessitating treatment approaches akin to those for urothelial carcinomas [25].

### 3.4. Molecularly Defined Renal Carcinomas

The 2022 WHO classification has formally established the category of “Molecularly Defined Renal Carcinomas,” marking a pivotal shift from a morphology-dominated to an integrated molecular–morphological framework [26]. This category encompasses several distinct entities, each with specific therapeutic implications.

TFE3-rearranged RCC is a rare kidney cancer subtype with early onset and poor prognosis. It arises from chromosomal translocations involving the Xp11.2 locus, leading to gene fusions between TFE3 and various partner genes—such as ASPSCR1, NONO and RBM10—which drive subsequent oncogenic molecular alterations [27]. In contrast, TFEB-altered RCC is subclassified into rearranged (often indolent) and amplified subtypes, the latter occurring more frequently in elderly patients, exhibiting aggressive behavior, and being associated with VEGFA co-amplification, suggesting potential susceptibility to VEGFR-targeted therapies [28]. FH-deficient RCC, frequently linked to hereditary leiomyomatosis, is driven by fumarate hydratase inactivation, leading to HIF accumulation and VEGF upregulation, which underscores its aggressive nature and potential responsiveness to specific strategies [29,30]. Other notable subtypes include SDH-deficient, ALK-rearranged (responsive to ALK inhibitors), ELOC (TCEB1)-mutated, and highly aggressive SMARCB1-deficient RCC [26]. The recognition of these molecularly defined entities provides a more precise pathological foundation for diagnosis, prognostication, and the selection of targeted therapies in clinical practice.

## 4. Treatment of nccRCC

For localized RCC, surgery is the primary treatment for both nccRCC and ccRCC. Historically, open radical nephrectomy, which has been the standard method, is now being reconsidered due to growing concerns about its negative impacts. As our understanding of tumors deepens, more surgical options have emerged, with the choice of surgical method now depending on tumor size and location [31].

For metastatic RCC, the outlook of nccRCC is usually poorer than ccRCC [32]. Patients with metastatic nccRCC tend to have shorter survival following cytoreductive nephrectomy compared to those with metastatic ccRCC [33]. Nevertheless, current management of metastatic nccRCC remains largely adapted from ccRCC, leading to a lack of subtype-specific evidence and imprecise treatment strategies. We summarize systemic therapies including chemotherapy, targeted agents, immunotherapy, and combinations to help guide future nccRCC treatment (Figure 3).

### 4.1. Cytotoxic Chemotherapy

Clinical trials evaluating the efficacy of cytotoxic chemotherapy in nccRCC have generally yielded unconvincing results. However, due to the distinct pathophysiological and genetic differences among histological subtypes, certain nccRCC subtypes, such as RMC and CDC, have demonstrated a more significant response to chemotherapy than ccRCC [34,35,36].

In a recent phase II multicenter, single-arm trial, sorafenib was administered in combination with gemcitabine and cisplatin to 26 treatment-naïve patients with metastatic CDC. The study documented an ORR of 30.8%, a median progression-free survival (PFS) of 8.8 months, and a mOS of 12.5 months [37]. In the therapeutic management of RMC, the recommended approach involves platinum-based chemotherapy regimens, typically in combination with either gemcitabine or paclitaxel [38,39]. Additionally, the use of bevacizumab combined with gemcitabine and platinum-based chemotherapy in 34 patients with metastatic CDC and RMC was evaluated in a phase II multicenter, single-arm study, which demonstrated an ORR of 41.2%, a mPFS of 5.9 months, and a mOS of 11.1 months [40]. However, the evidence supporting chemotherapy in CDC and RMC originates almost exclusively from non-randomized, single-arm studies. Consequently, the reported efficacy outcomes, such as the ORR of 30.8–41.2%, must be viewed as preliminary. These estimates are vulnerable to bias from patient selection and the lack of a comparator arm. Although cytotoxic chemotherapy retains a role in specific nccRCC subtypes, it is largely ineffective in the majority of cases. Consequently, targeted therapy and immunotherapy have emerged as the principal treatment modalities for most forms of this disease.

### 4.2. Targeted Therapies

#### 4.2.1. mTOR Inhibitors

Excessive activation of mTOR is sometimes observed in nccRCCs. Its hyperactivation drives tumorigenesis by upregulating HIF-1α, which subsequently transactivates key pro-angiogenic factors like VEGF, PDGF, and GLUT1, all of which contribute to tumor progression [41]. Studies have demonstrated that inhibiting mTOR exerts antitumor effects in nccRCC, highlighting its potential as a therapeutic target [42]. Among systemic options for nccRCC, the mTOR-inhibitor class is essentially anchored by temsirolimus and everolimus.

Everolimus has exhibited some therapeutic potential in nccRCC. In a phase II multicenter trial involving 49 patients with nccRCC treated with everolimus monotherapy, the mPFS was observed to be 5.2 months, while the mOS reached 14.0 months. Notably, patients with chRCC demonstrated a more favorable PFS outcome compared to those with other histological subtypes [43]. However, results from the NCT01120249 trial demonstrated that adjuvant everolimus did not improve recurrence-free or OS in patients with papillary or chromophobe RCC [44]. These findings reflect the limited efficacy of mTOR inhibitors in high-risk settings, highlighting the urgent need for more effective therapeutic approaches.

#### 4.2.2. MET Inhibitors

The MET proto-oncogene, located on chromosome 7q31, is closely linked to the development of pRCC, particularly type 1 pRCC [45]. As a receptor tyrosine kinase (RTK), the MET gene can become overactivated through various mechanisms, including somatic mutations, alternative splicing, and gene fusions. Upon activation, the MET signaling pathway contributes to tumorigenesis by enhancing tumor cell proliferation, invasion, and angiogenesis [11,46]. In the systemic management of nccRCC, the MET-directed therapeutic repertoire most commonly comprises foretinib, savolitinib, crizotinib and cabozantinib.

Clinical evidence supports the efficacy of MET inhibitors specifically in MET-driven pRCC. In a phase II multicenter trial involving 74 patients with pRCC, foretinib, the first MET-targeted agent used in nccRCC achieved an overall ORR of 13.5%. However, the response rate was substantially higher in MET-driven tumors (50%) compared to MET-independent cases (9%), with a mPFS of 9.3 months [47]. Cabozantinib has demonstrated superior efficacy in patients with MET-altered RCC. Notably, in the phase II CABOSUN trial, cabozantinib significantly prolonged mPFS compared to sunitinib, achieving 13.8 months vs. 3.0 months in the MET-driven subgroup [48]. A retrospective study of 112 patients with advanced nccRCC further supported this, showing a 40% partial responses (PRs) with cabozantinib in MET-altered pRCC [49]. Crucially, the phase II SWOG 1500 trial directly compared MET inhibitors in metastatic pRCC: cabozantinib outperformed sunitinib in both median PFS (9.0 vs. 5.6 months) and ORR (23% vs. 4%), while crizotinib and savolitinib did not show similar superiority [6]. The collective evidence solidifies the role of MET inhibition in pRCC, yet it also reveals a hierarchy of efficacy among agents. The superior and consistent performance of cabozantinib, as demonstrated in the randomized SWOG 1500 trial, suggests that its multi-kinase inhibition profile (VEGF/MET) may confer a broader antitumor effect compared to selective MET inhibitors like savolitinib and crizotinib, which showed more variable outcomes across studies. These findings highlight the heterogeneity among MET-targeting strategies, thereby solidifying the position of cabozantinib as the preferred therapeutic option for MET-altered pRCC. Furthermore, the marked superiority of treatment efficacy in MET-driven over non-MET-driven tumors underscores the critical importance of implementing molecularly guided, personalized approaches for nccRCC.

#### 4.2.3. Multitarget Tyrosine Kinase Inhibitors

RTKs such as MET, KIT, PDGFR, VEGFR, and FGFR orchestrate key oncogenic processes in RCC, including proliferation, angiogenesis, and invasion [50,51]. Given their frequent upregulation, multitargeted TKIs (e.g., Sunitinib, Cabozantinib) that concurrently inhibit VEGFR, PDGFR, and MET offer clinical advantages. By blocking multiple tumor-promoting pathways, these agents reduce the risk of resistance arising from single-target inhibition and provide broader coverage against tumor heterogeneity or compensatory signaling, thereby underpinning their broad application in nccRCC.

Sunitinib, a classical multitarget TKI, exerts its effects by inhibiting critical signaling pathways, including VEGFR, PDGFR, and other RTKs [52]. Although systematic reviews indicate a lower evidence level for sunitinib in nccRCC compared to ccRCC, clinical trials reveal significant subtype-dependent responses [53]. In a phase II trial (Tannir et al.), sunitinib monotherapy yielded a mPFS of only 1.6 months in pRCC vs. 12.7 months in chRCC [54]. Although its efficacy varies across nccRCC subtypes, sunitinib demonstrates comparative advantage over mTOR inhibitors, as evidenced by randomized controlled trials. In the ASPEN trial, sunitinib significantly prolonged mPFS compared to everolimus (8.3 vs. 5.6 months) [55]. This finding was corroborated by the ESPN trial, which reported not only a longer mPFS (6.1 vs. 4.1 months) but also a significant OS benefit with sunitinib (31.6 vs. 10.5 months) [56].

Cabozantinib, a multi-target TKI inhibiting MET, VEGFR2, and others, demonstrates broad anti-tumor activity across nccRCC subtypes [57,58]. Retrospective analyses have consistently demonstrated the efficacy of this therapeutic approach across multiple studies. A recent study from Martínez Chanzá et al., which included 112 metastatic nccRCC patients reported an ORR of 27%, with median PFS and OS of 7.0 months and 12.0 months, respectively. Importantly, the treatment exhibited a manageable safety profile that the incidence of grade 3 or higher toxicities was low, occurring in no more than 4% of patients [49]. Activity has also been observed in rare nccRCC subtypes. The BONSAI trial, which enrolled 25 patients with untreated metastatic CDC, met its primary endpoint by demonstrating an ORR of 35%. The regimen was well-tolerated with no permanent treatment discontinuations, and resulted in a mPFS of 4 months and a mOS of 7 months [59]. Similarly, cabozantinib showed clinically meaningful efficacy in 52 patients with metastatic MiT family RCC, yielding an ORR of 17.3%, a mPFS of 6.8 months, and a mOS of 18.3 months. These outcomes suggest superior activity compared to historical data from VEGFR-TKI therapies [60]. However, the promising efficacy signals, particularly in rare subtypes like CDC and MiT family RCC, are derived from small cohorts, necessitating validation in larger, prospective studies to definitively establish its role relative to other agents.

Axitinib, pazopanib, and lenvatinib have also demonstrated potential in the treatment of nccRCC. As a potent inhibitor of VEGFR, PDGFR, and KIT, axitinib resulted in a mPFS of 6.6 months, a mOS of 18.9 months, and an ORR of 28.6%, indicating clinically meaningful benefit in a phase II trial involving 44 treatment-naïve patients with metastatic pRCC [61]. Pazopanib, which exerts its antitumor effects by inhibiting VEGFR, PDGFR, and KIT, represents a standard first-line treatment for ccRCC and has also demonstrated clinical activity in nccRCC [62,63], which yields ORR of 27% to 33%, disease control rates (DCR) of 81% to 89%, mPFS ranging from 8.1 to 16.5 months, and OS between 17.3 and 31.0 months [64]. For advanced or refractory cases, the combination of lenvatinib and everolimus has shown promising efficacy. A phase II study involving 31 patients reported an ORR of 26%, mPFS of 9.2 months, and mOS of 15.6 months. Notably, enhanced responses were observed in patients with chromophobe histology, who achieved an ORR of 44%, a finding consistent with a supporting case report [65,66].

#### 4.2.4. Other Targeted Molecular Therapies

Bevacizumab is a humanized monoclonal antibody directed against VEGF [67]. While bevacizumab monotherapy lacks supporting data in nccRCC, combination regimens show variable efficacy. Bevacizumab plus everolimus demonstrated significant activity in papillary variant RCC. Trials combining these agents (*n* = 35, *n* = 39) reported mPFS of 11.0–13.7 months, mOS of 18.5–33.9 months, and ORR of 29–35%, with responses notably higher in papillary or chromophobe subtypes [68,69].

Erlotinib, a TKI targeting EGFR, has demonstrated therapeutic activity in advanced or metastatic pRCC patients in a Phase II trial, which showed an ORR of 11% and a mOS of 27 months [70]. A Phase II trial by Srinivasan et al. in patients with HLRCC, where erlotinib combined with bevacizumab, resulted in an ORR of 51% and a mPFS of 14.2 months, alongside manageable toxicity, suggesting erlotinib-based combination therapies to have promising prospect and need further investigation [71].

### 4.3. Immune Checkpoint Inhibitors

While current clinical guidelines favor TKIs for metastatic nccRCC, resistance across diverse nccRCC subtypes has spurred exploration of novel therapies, particularly immune checkpoint inhibitors (ICIs) [72]. ICIs targeting CTLA-4 and PD-1/PD-L1 have largely supplanted older immunotherapies like interferons/interleukins due to superior efficacy [73]. Their innovation lies in overcoming dual T-cell suppression: CTLA-4 inhibitors enhance T-cell activation in lymph nodes by blocking inhibitory signals during immune priming, while PD-1/PD-L1 inhibitors counteract T-cell exhaustion within the tumor microenvironment, restoring cytotoxic function [74,75,76]. This augmented immune response improves patient survival and has transformed nccRCC treatment. ICI therapy holds particular clinical significance for PD-L1-high nccRCC subtypes, which are associated with worse outcomes and higher tumor stages [77]. Currently, the ICIs that are widely used in nccRCC mainly include Nivolumab, Pembrolizumab and Tremelimumab.

Nivolumab, a fully human anti-PD-1 monoclonal antibody, shows variable but clinically meaningful activity in metastatic nccRCC [78]. Retrospective analyses have consistently reported modest outcomes. Koshkin et al., in a study of 41 patients with predominant papillary (39%) and unclassified (34%) histology, documented an ORR of 20%, a DCR of 49%, and a mPFS of 3.5 months [79]. These findings were corroborated by a trial from Chahoud et al. (*n* = 40), which demonstrated an ORR of 20.6%, a DCR of 70.5%, a mPFS of 4.9 months, and a mOS of 21.7 months. Notably, efficacy varied significantly by subtype: unclassified RCC and ccRCC with rhabdoid features exhibited higher response rates (44.4% and 28.6%, respectively), whereas type 2 papillary, chromophobe, and translocation RCC subtypes showed minimal responses [80]. This pattern of subtype-dependent activity is further supported by prospective data from the CheckMate 374 trial (*n* = 44), which reported an ORR of 13.6%, a mPFS of 2.2 months, and a mOS of 16.3 months [81]. These findings highlight both the promise and limitations of PD-1 inhibition in nccRCC. While responses in specific subtypes are encouraging, the current data, predominantly from retrospective cohorts, lack the statistical power and prospective validation to definitively associate specific subtypes with nivolumab response. The consistently modest PFS across studies and the profound heterogeneity in response underscore the current challenge: the absence of validated biomarkers to guide patient selection beyond histology remains a major barrier to optimizing immunotherapy in this diverse disease.

Pembrolizumab demonstrated clinically significant activity in untreated advanced nccRCC in the KEYNOTE-427 cohort B (*n* = 165), the largest prospective monotherapy trial to date. The overall ORR was 26.7%, with notable subtype heterogeneity: unclassified (30.8%), papillary (28.8%), and chromophobe (9.5%). PD-L1 expression, defined as a combined positive score of 1 or higher, was associated with an enhanced response: these patients achieved an ORR of 35.3%, along with a mPFS of 5.6 months, mOS of 30.0 months, and a DCR of 50% [82]. This study provides the first prospective evidence supporting pembrolizumab monotherapy as a viable first-line option for nccRCC, demonstrating durable antitumor activity and manageable safety, particularly in PD-L1-positive patients and those with papillary/unclassified histology. However, the single-arm design, disease heterogeneity, and limited sample sizes for certain subtypes constrain the generalizability and comparative interpretation of these findings.

In contrast, tremelimumab (an IgG2 monoclonal antibody directed against CTLA-4) showed limited efficacy in a smaller nccRCC cohort (*n* = 11), yielding mPFS of 3.0 months and OS of 16.2 months regardless of monotherapy (*n* = 9) or cryoablation combination (*n* = 2) [83]. The consistently poor outcomes with CTLA-4 inhibition across these studies suggest that nccRCC may be inherently resistant to this immunotherapeutic approach, possibly requiring combination strategies to overcome this limitation.

Given the restricted effectiveness of single-agent ICIs, a multitude of research endeavors have delved into ICI combination regimens tailored for advanced nccRCC patients. The most commonly explored pairing is Nivolumab in conjunction with Ipilimumab(a human CTLA-4 antibody) [84]. In the single-arm, phase IIIb/IV CheckMate 920 study, 52 treatment-naive patients with advanced nccRCC received nivolumab plus ipilimumab, yielding an ORR of 19.6%, a mPFS of 3.7 months and a mOS of 21.2 months [8]. In addition, the difference in the effect of combined immunotherapy is strongly related to different subtypes of nccRCC. A recent study in 55 nccRCC patients confirmed this variability: ORR was 36.4% overall, highest in pRCC (ORR 48%, mPFS 10.6m, mOS 36.7m), and lower in chRCC and unclassified RCC. Sarcomatoid features correlated with higher ORR. Molecular profiling revealed frequent alterations in TP53 (42%), PTEN (23%), and TERT (23%). Notably, these mutations were associated with differential responses to nivolumab plus ipilimumab. Specific molecular alterations, such as TERT and SETD2 mutations, were associated with objective responses to nivolumab-ipilimumab, whereas TP53 alterations were enriched in the less responsive chromophobe subtype. These findings underscore the potential predictive value of molecular profiling for ICI efficacy in nccRCC [85].

### 4.4. Combination Strategies of TKI and ICI

TKI therapy improves vascular conditions and reduces immunosuppressive molecule expression on tumor endothelia while modulating immune cell function. Synergizing with ICI, it enhances T-cell infiltration and activity, overcoming immunosuppressive microenvironments to boost immunotherapy efficacy [86]. Clinically, this synergy is reflected in the superior outcomes of ICI-TKI combinations over TKI monotherapy across different treatment lines in advanced nccRCC.

Clinical evidence continues to support the synergistic potential of combination therapy in nccRCC. In a phase Ib trial by McGregor et al., the combination of cabozantinib and atezolizumab yielded an ORR of 47% in papillary, 11% in chromophobe, and 25% in other subtypes, alongside a mPFS of 9.5 months and a DCR of 94% among 32 patients [87]. Similarly, a phase II trial evaluating cabozantinib plus nivolumab in 47 patients reported an ORR of 47.5% and mPFS of 12.5 months in those with papillary, FH-deficient, translocation, or unclassified RCC, along with a mOS of 28 months. By contrast, no response was observed in chromophobe subtypes, indicating preferential efficacy in papillary-featured variants [88]. Updated results from this cohort further confirmed a 48% ORR, mPFS of 13 months, and sustained survival rates of 70% at 18 months and 44% at 36 months [89].

Additional evidence supports the efficacy of various TKI-ICI combinations across nccRCC subtypes. The NEMESIA trial, which enrolled 32 patients and evaluated pembrolizumab plus axitinib, demonstrated an ORR of 43.7%, a DCR of 78.1%, and a mPFS of 10.8 months. Responses varied by subtype, with ORRs of 47.3% in papillary and 41.6% in chromophobe histology [90]. Further supporting these findings, the KEYNOTE-B61 trial investigated pembrolizumab combined with lenvatinib in 158 patients and achieved an ORR of 49%, a mPFS of 18 months, and a mOS that had not yet been reached, with a 12-month OS rate of 82% [91].

Subtype-focused analysis reveals MET-driven pRCC responds robustly to savolitinib and durvalumab (ORR: 53%, mPFS: 12.0 months) [92]. Emerging combinations include tislelizumab and TKI, yielding a 40% response rate and 11.9-month PFS, figures that compare favorably with 10.5% and 4.6 months observed when the same TKI is administered as monotherapy [93,94]. These findings collectively reinforce TKI-ICI combinations as clinically valuable strategies (Table 1).

Collectively, these findings establish TKI-ICI combinations as valuable strategies in nccRCC, yet the current evidence base—dominated by single-arm trials—inherently limits comparative assessments. The marked efficacy variations across subtypes highlight the inadequacy of a uniform approach, while the success of biomarker-driven therapy in MET-altered pRCC underscores the need for molecularly defined patient selection. Future trials incorporating comprehensive biomarker evaluation will be essential to optimize combination strategies and validate predictive biomarkers.

Meanwhile, given the marked heterogeneity of nccRCC and its relatively low incidence, a universally applicable treatment paradigm remains elusive. Current management often extrapolates from experiences with ccRCC, primarily employing targeted therapy and immunotherapy. However, the profound heterogeneity of nccRCC subtypes, characterized by distinct tumor microenvironments (TMEs), metabolic profiles, and molecular drivers, confers significantly divergent biological behaviors and therapeutic responses. A subtype-specific management approach (Figure 2). For instance, identifying MET alterations in pRCC has enabled targeted inhibitors that outperform broad-spectrum therapies [95]. In this context, liquid biopsy—particularly circulating tumor DNA (ctDNA) profiling—has emerged as a promising, non-invasive tool for diagnosing and managing nccRCC [96]. Despite molecular heterogeneity and low tumor shedding in certain subtypes, advanced techniques such as targeted NGS and epigenomic analyses like cell-free methylated DNA immunoprecipitation and sequencing (cfMeDIP-seq) enable highly sensitive detection of molecular alterations and subtype discrimination [97].

However, the historical and still prevalent practice of grouping all nccRCCs in clinical trials has been driven by pragmatic challenges, including the rarity of individual subtypes and previously limited diagnostic capabilities [98]. While this aggregated approach has yielded general treatment frameworks, it inevitably obscures efficacy for specific histologies.

To address the critical gap and move beyond the limitations of historical trials that aggregated all subtypes, contemporary research is increasingly focused on two key strategies: evaluating combination therapies—notably TKIs with ICIs—in broad nccRCC cohorts, and prospectively investigating novel agents within specific histologic or molecularly defined subtypes (Table 2).

## 5. Molecular Subtypes Based on Multi-Omics

Given the extreme heterogeneity of nccRCC, regarding it as a uniform group or relying solely on histological classification no longer satisfies the requirements of precision medicine. The majority of treatment approaches for nccRCC are extrapolated from ccRCC, leading to limited nccRCC-specific evidence. Nevertheless, advancements of technologies such as NGS and single-cell RNA sequencing (scRNA-seq) have deepened molecular insights, paving the way for a paradigm shift towards molecular-subtype-driven treatments.

Tumor heterogeneity encompasses both intra- and inter-tumoral diversity. While early classification systems focused predominantly on cancer cells themselves, emerging evidence highlights substantial heterogeneity within the TME—particularly among infiltrating immune cells and stromal components. These elements critically shape immunosuppressive or supportive niches, directly impacting therapeutic outcomes such as response to immunotherapy or the emergence of drug resistance [99]. Therefore, molecular subtyping that integrates multi-dimensional data—from tumor cells to the TME—offers a more clinically informative framework, overcoming the limitations of histology-only classifications and enabling more precise treatment strategies.

Through single-omics profiling, studies have delineated the molecular landscape of nccRCC to inform subtype classification and potential therapies. One investigation integrated nccRCC scRNA-seq with TCGA bulk RNA-seq data, applying CIBERSORTx to deconvolute the TME and identify three distinct subtypes: an abundance of exhausted CD8^+^ T cells, tumor-associated macrophages (TAMs) derived from chRCC, and sarcomatoid carcinoma cells. Further analysis of the peritumoral microenvironment suggested that drugs targeting key molecules such as HAVCR2 and TREM2 in this high-risk subtype might hold therapeutic potential [100]. Another transcriptomic analysis revealed significant pathway activity differences among RCC subtypes—e.g., elevated FAO and AMPK in chRCC versus increased FAS and pentose phosphate pathway in pRCC. This study also confirmed greater infiltration of B cells, fibroblasts, neutrophils, and monocytic cells in nccRCC compared to ccRCC, underscoring considerable inter-subtype TME heterogeneity [101]. Collectively, these single-omics analyses provide a molecular basis for subtyping nccRCC and highlight actionable targets for tailored treatment.

Though single-omics analysis has proven effective for depicting intratumoral and intertumoral heterogeneity in nccRCC, the continuous advancement of multi-omics techniques (encompassing genomics, transcriptomics, epigenomics, proteomics, and metabolomics) has spurred the expanding use of multi-omics integration in cancer categorization. Studies have shown that cancer subtyping based on multi-omics is more dependable and efficient than analysis grounded in single-omics or particular molecular signatures [102]. Notably, multi-omics analysis possesses significant potential for in-depth analysis of the TME. Its application in immune microenvironment research, which includes probing immune cell infiltration patterns, evaluating immune checkpoint molecule expression, and analyzing cytokine networks, is critical for comprehending tumor immune evasion and forecasting immunotherapy responses [103].

Current multi-omics molecular subtyping research in RCC primarily centers on ccRCC. For instance, one study conducted WES, whole-transcriptome sequencing (WTS), proteomics, metabolomics, and spatial transcriptomics on tumor samples from 100 ccRCC patients and 50 paired normal adjacent tissues (NATs). Through this multi-omics analysis, a novel ccRCC subtype termed DCCD-ccRCC was defined. Moreover, via scRNA-seq and spatial trajectory analysis, ccRCC was stratified into four immune subtypes. Among them, IM4 (DCCD) tumors benefit from anti-VEGF + ICI combination, whereas IM2 tumors achieve favorable outcomes with TKI monotherapy [104]. In 2019, a comprehensive multi-omics analysis of 103 treatment-naive ccRCC samples—encompassing genomics, epigenomics, transcriptomics, proteomics, and phosphoproteomics—employed transcriptomic deconvolution via the xCell tool to define four immune-based subtypes linked to genomic alterations and tumor microenvironment features. The CD8+ Inflamed subtype, enriched with immune checkpoint markers and BAP1 mutations, may respond to ICIs. The CD8- Inflamed and VEGF Immune Desert subtypes, showing angiogenic signatures, are candidates for anti-VEGF therapy. The Metabolic Immune Desert subtype, characterized by metabolic activation, may benefit from mTOR-targeted approaches, enabling rational treatment stratification. In addition, unsupervised proteomic clustering further identified three major subgroups associated with tumor grade, stage, and specific mutations, offering critical molecular insights and highlighting potential therapeutic targets [105].

Compared to ccRCC, nccRCC lacks a well-developed multi-omics based molecular classification. This is due to its marked tumor-to-tumor and within-tumor heterogeneity, as well as its lower incidence. In a study of 823 RCC tumors designed to link molecular subtypes to the treatment efficacy of tyrosine kinase inhibitors and immune checkpoint inhibitors, RNA-seq coupled with unsupervised non-negative matrix factorization (NMF) uncovered seven RCC subtypes. It is worth noting that nccRCC with sarcomatoid features mostly clustered into the proliferative/stromal subtype (Cluster 5–6), and all tumors with TFE3/TFEB fusions were grouped into the high-proliferation subtype (Cluster 5) [106]. Another multi-omics study on 113 RCC patients used proteomics to classify RCC into four subtypes (C1–C4). C4 was composed nearly entirely of nccRCC and was characterized by active mitochondrial and oxidative phosphorylation pathways [107]. Although a refined molecular classification capturing the internal heterogeneity of nccRCC remains to be established, these studies provide preliminary insights into nccRCC biology and targeted therapy strategies. Significant therapeutic heterogeneity was observed across molecular clusters: the atezolizumab+ bevacizumab combination outperformed sunitinib in clusters 4, 5, and 7 (HR 0.45–0.62, *p* < 0.01), while clusters 1 and 2 showed comparable efficacy between regimens. Clusters 3 and 6 exhibited intermediate responses with modest effect sizes (HR 0.78–0.85, *p* = NS). The results derived from this study have been successfully implemented into clinical practice. Specifically, these findings are now informing the design and execution of an ongoing Phase II clinical trial, thereby contributing valuable evidence to support treatment decision-making for affected patients.

So far, significant progress in the multi-omics molecular classification of nccRCC has been made by Li and colleagues [107]. Through a comprehensive proteogenomic characterization of 48 nccRCC samples, the study employed immune deconvolution algorithms to infer the proportion of immune cells in the tumor microenvironment. Cluster analysis divided the samples into three immune infiltration subtypes: myeloid–lymphoid high-infiltration (immune-active tumors), myeloid high-infiltration (mainly pRCC), and immune-depleted (encompassing renal oncocytomas and chRCC). Compared to ccRCC, this classification not only reveals the low immune infiltration in nccRCC but also shows that the myeloid-lymphoid high-infiltration subtype has both high immune infiltration and high genomic instability. This classification not only underscores generally lower immune infiltration in nccRCC relative to ccRCC, but also links the myeloid–lymphoid high-infiltration group with heightened genomic instability—a connection further supported by glycoproteomic profiling. This study delineates microenvironment-driven subtypes tied to genomic and post-translational features, offering a basis for tailored immunotherapy. For instance, the myeloid–lymphoid subtype may respond better to immune checkpoint inhibition, while immune-depleted cases demand alternative strategies. Furthermore, integration of RNA, protein, and phosphorylation data via ARD–NMF categorized 151 samples into six multi-omics clusters. Clusters ARD–NMF-0 (pRCC-enriched) and ARD–NMF-3 (renal oncocytomas) showed more favorable prognosis compared to ARD–NMF-1 (ccRCC-rich, high methylation, poor outcome).

Despite its promise, multi-omics subtyping in nccRCC faces considerable challenges. A primary limitation is the restricted cohort size, largely due to the rarity of nccRCC and its individual subtypes, which often results in underpowered analyses and an inability to fully capture intrinsic heterogeneity. With some subtypes comprising only one or two samples, the available data is inadequate to capture the heterogeneity within them. Furthermore, the absence of standardized procedures presents a major barrier to data integration, preventing the subtyping results from being reliably repeated and widely applicable. Despite these hurdles, multi-omics profiling remains crucial for deciphering nccRCC biology and holds considerable promise for paving the way to tailored therapeutic strategies.

While multi-omics profiling has provided frameworks for molecular subtyping in RCC, the correlation of these subtypes with drug sensitivity and therapeutic responses is an area that requires further exploration. In a 2018 study, a biobank of bladder cancer organoids was established that recapitulated tumor heterogeneity and enabled systematic pharmacodynamic analysis, providing a robust platform for personalized urothelial cancer treatment. Analyses of drug response utilizing bladder tumor organoids reveal partial correlations with mutational profiles and changes linked to treatment resistance. Specific responses can be validated through xenografts in vivo [108]. Similarly, in hepatocellular carcinoma, organoid biobanks derived from clinical specimens have revealed genomic and transcriptomic markers associated with drug response through high-throughput screening of seven therapeutic agents [109]. These studies indicate that patient-derived tumor organoids represent a faithful model system for studying tumor evolution and treatment response in the context of precision cancer medicine. In contrast, such integrative approaches are still lacking in RCC. In particular, the integration of organoid-based drug testing with comprehensive multi-omics profiling presents a significant opportunity to enhance precision oncology in the context of this malignancy.

## 6. Conclusions

NccRCC represents a molecularly and histologically diverse group of kidney cancers. The 2022 WHO classification underscores the transition towards molecularly defined entities, yet optimal treatment strategies for advanced disease remain challenging, often relying on extrapolations from ccRCC. Molecular characterization and multi-omics subtyping are crucial for refining diagnosis, prognostication, and personalizing therapy with targeted agents and immunotherapies.

As our comprehension of omics analysis and the TME continues to deepen, an innovative molecular classification approach grounded in multi-omics technology has emerged. By taking into account both the inter-tumoral heterogeneity and the unique characteristics of the TME, this method can accurately classify nccRCC, thereby offering a clearer path towards precision medicine. For nccRCC, due to the rarity of samples and the difficulty in pathological identification, large-scale samples to obtain comprehensive multi-omics information is currently lacking, which has significantly impeded the progress of its molecular classification research. Recent progress in biotechnology has led to the emergence and refinement of various in vitro models, establishing the foundation for the study of tumor molecular classification, drug sensitivity, and the achievement of individualized treatment [110]. A study on colorectal cancer organoids uncovered tumor cell heterogeneity and intercellular interactions via scRNA-seq, deepening the understanding of molecular classification [111]. Thus, organoid models might provide robust support for future individualized treatment of nccRCC.

Despite progress, the current evidence base is largely derived from retrospective studies and small prospective trials. Future efforts need to focus on validating molecular subtypes in larger cohorts to ultimately improve clinical outcomes for patients with nccRCC.

## Figures and Tables

**Figure 1 cells-14-01781-f001:**
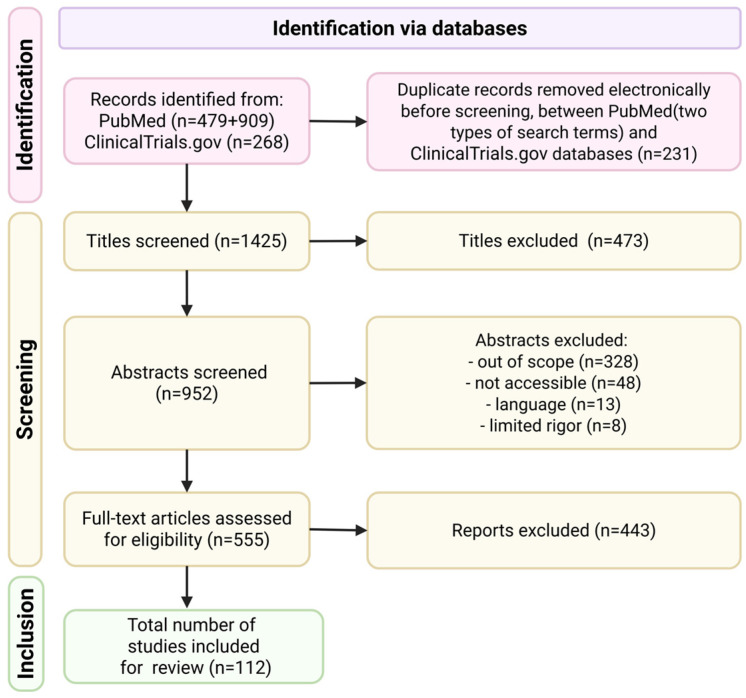
Search procedures and results.

**Figure 2 cells-14-01781-f002:**
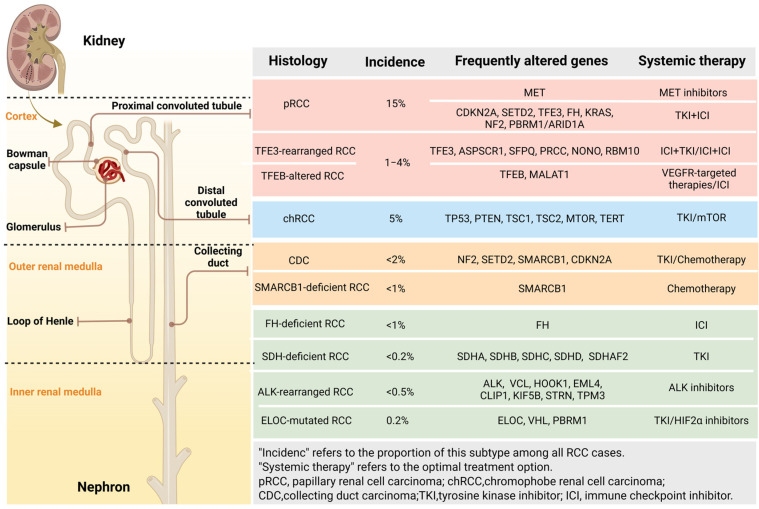
Comprehensive overview of the clinicopathological and molecular characteristics of nccRCC.

**Figure 3 cells-14-01781-f003:**
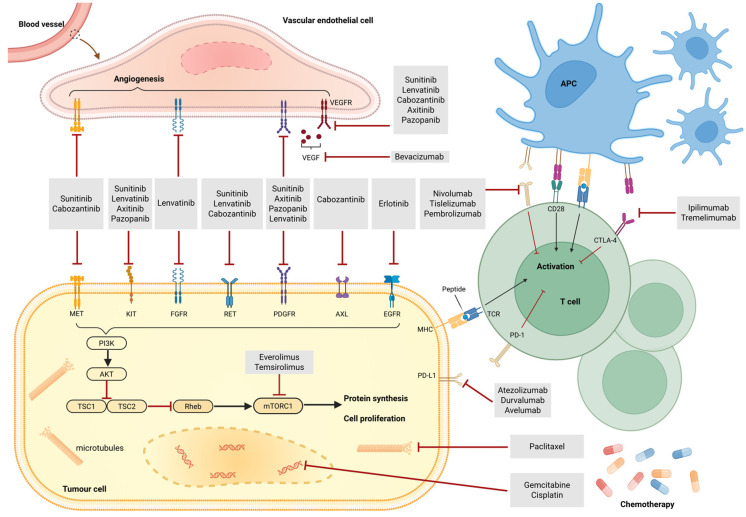
The systemic treatment plan of nccRCC.

**Table 1 cells-14-01781-t001:** Results from retrospective studies and clinical trials in patients with metastatic nccRCC.

Study/Author Name	PMID	Phase	Histology	No. of Patients	Treatments	ORR (%)	mPFS (Months)	mOS (Months)
NCT01762150 [37]	29933095	phase II	mCDC	26	Sorafenib + Gemcitabine + Cisplatin	30.8	8.8	12.5
BEVABEL-GETUG/AFU24 [40]	37054556	phase II	mCDC RMC	31 3	Gemcitabine + Bevacizumab + Cisplatin (Carboplatin)	41.2	5.9	11.1
NCT00830895 [43]	23180114	phase II	nccRCC	49	Everolimus	10.2	5.2	14.0
NCT00726323 [47]	23213094	phase II	pRCC	74	Foretinib	13.5	9.3	NR
SWOG 1500 [6]	33592176	randomized phase II	pRCC	152	Cabozantinib vs. Sunitinib	23 (cabozantinib)	9.0 (cabozantinib)	20 (cabozantinib)
					vs. Crizotinib vs. Savolitinib (halted)	4 (sunitinib)	5.6 (sunitinib)	16.4 (sunitinib)
Tannir [54]	22771265	phase II	nccRCC	57	Sunitinib	5	2.7	16.8
ASPEN [55]	26794930	randomized phase II	nccRCC	108	Sunitinib vs. Everolimus	18 vs. 8	8.3 vs. 5.6	31.5 vs. 13.2
ESPN [56]	26626617	randomized phase II	nccRCC	56	Sunitinib vs. Everolimus	11 vs. 3	6.1 vs. 4.1	16.2 vs. 14.9
NCT01835158 [48]	29550566	randomized phase II	RCC	157	Cabozantinib vs. Sunitinib	20 vs. 9	8.6 vs. 5.3	26.6 vs. 21.2
NCT02982954 [49]	30827746	retrospective	nccRCC	112	Cabozantinib	27	7.0	12
BONSAI [59]	35420628	phase II	mCDC	23	Cabozantinib	35	4	7
Thouvenin [60]	35979929	retrospective	tRCC	52	Cabozantinib	17.3	6.8	18.3
AXIPAP [61]	32146304	phase II	pRCC	44	Axitinib	28.6	6.6	18.9
Matrana [62]	27568124	retrospective	nccRCC	29	Pazopanib	33 (Frontline)	8.1 (Frontline)	31 (Frontline)
						6 (secondline)	4 (secondline)	13.6 (secondline)
Jung [63]	28546525	phase II	nccRCC	29	Pazopanib	28	16.5	NR
CREATE [66]	33867192	phase II	nccRCC	31	Lenvatinib + Everolimus	26	9.2	15.6
Voss [69]	27601542	phase II	nccRCC	34	Bevacizumab + Everolimus	29	11	18.5
NCT01130519 [71]		prospective	pRCC	83	Bevacizumab + Erlotinib	51	14.2	NR
SWOG S0317 [70]	19884559	phase II	pRCC	45	Erlotinib	11	NE	27
Koshkin [79]	29378660	retrospective	nccRCC	41	Nivolumab	20	3.5	NR
CheckMate 374 [81]	32718906	Phase IIIb/IV	nccRCC	44	Nivolumab	13.6	2.2	16.3
KEYNOTE 427 [82]	33529058	phase II	nccRCC	165	Pembrolizumab	26.7	4.2	28.9
NCT02626130 [83]	34737281	pilot study	nccRCC	11	Tremelimumab + Cryoablation	NR	3.0	22.7
					vs. Tremelimumab		vs. 5.0	vs. 33.7
CheckMate 920 [8]	35210307	phase IIIb/IV	nccRCC	52	Nivolumab + Ipilimumab+	19.6	3.7	21.2
			RMC	3	Platinum-based chemotherapy			
Moussa [85]	39939142	retrospective	pRCC	25	Nivolumab + Ipilimumab	48	10.6	36.7
			chRCC	12		25	3.6	25.7
			uRCC	18		27.8	3	11.1
COSMIC-021 [87]	34491815	Phase Ib	nccRCC	32	Cabozantinib + Atezolizumab	31	9.5	NR
Lee [88]	35298296	phase II	nccRCC	40	Cabozantinib + Nivolumab	47.5	12.5	28
NEMESIA [90]	36674615	observational	nccRCC	32	Pembrolizumab + Axitinib	43.7	10.8	NR
KEYNOTE-B61 [91]	37451291	phase II	nccRCC	158	Pembrolizumab + Lenvatinib	49	18	NR
CALYPSO32 [92]	36809050	phase II	pRCC	41	Savolitinib + Durvalumab	29	4.9	14.1
			MET-driven	27		53	12.0	27.4
Nonomura [93]	39699015	observational	nccRCC	22	Avelumab + Axitinib	22.7	7.5	NR
Zhong [94]	39903308	retrospective	nccRCC	39	Tislelizumab + TKI	40	11.9	NR

Abbreviations: nccRCC, non-clear cell renal cell carcinoma; chRCC, chromophobe RCC; NR, not reached; ORR, objective response rate; mOS, median overall survival; mPFS, median progression-free survival; pRCC, papillary RCC; tRCC, translocation RCC; RMC, renal medullary carcinoma. All efficacy endpoints (ORR, DCR, PFS, OS) are assessed per RECIST 1.1 unless otherwise specified.

**Table 2 cells-14-01781-t002:** Selected ongoing clinical trials.

Clinical Trial	Study Design	Enrollment	Histology	Treatment	Primary Endpoint
NCT06053658	phase II	48	nccRCC	Tivozanib + Nivolumab	Safety
NCT05831891	phase II	39	nccRCC	Fruquintinib + Serplulimab	PFS
FRONTIER					
NCT05678673	randomized	317	nccRCC	Sunitinib vs. XL092 + Nivolumab	PFS, ORR
STELLAR-304	phase III				
NCT04413123	phase II	60	nccRCC	Cabozantinib + Nivolumab + Ipilimumab	ORR
NCT05808608	phase I/II	33	ssRCC	AK104 + Axitinib	ORR
NCT06302569	phase II	23	CDC	Pembrolizumab + enfortumab	ORR
REPRINT			RMC		
NCT05411081	randomized	200	pRCC	Cabozantinib vs.	PFS
PAPMET2	phase II			Atezolizumab + cabozantinib	

Abbreviations: nccRCC, non-clear cell renal cell carcinoma; ORR, objective response rate; PFS, progression-free survival; pRCC, papillary RCC; RMC, renal medullary carcinoma; CDC, collecting duct carcinoma; ssRCC, special pathological subtypes of renal cell carcinoma; AK104, Anti-PD-1/CTLA-4 bi-specific antibody drug; XL092, Zanzalintinib.

## Data Availability

No new data were created or analyzed in this study.

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
