# Peer review of "Recent Advances in nccRCC Classification and Therapeutic Approaches"

_cells, 2025, doi:10.3390/cells14221781_

Round 1

Reviewer 1 Report

Comments and Suggestions for Authors

The manuscript provides a comprehensive narrative review that is focused on the classification, molecular background and therapeutic management of non-clear cell renal cell carcinoma (nccRCC). The topic is relevant and timely. This is particularly the case in light of the increasing clinical importance of molecularly guided therapies. The authors address epidemiology. They also address molecular subtypes. And they address systemic treatment options (tyrosine kinase inhibitors, mTOR inhibitors, MET inhibitors). The authors also address future directions. These include multi-omics approaches and organoid models. The author has organised the manuscript well and supplemented it with informative figures and a table that effectively summarises the current therapeutic landscape.

Before the review can be published, several methodological and factual issues need to be addressed, despite its clinical value. The review methodology is not clearly described in the manuscript, which makes it unclear how the literature was searched, selected, and analysed. Including a 'Methods' section detailing the databases used (e.g. PubMed, Embase and ClinicalTrials.gov), the search terms used, the inclusion and exclusion criteria applied, the date of the last search and whether abstracts or non-English papers were considered, would improve transparency and reduce the risk of selection bias. Incorporating a PRISMA-style flow diagram would further enhance the review's credibility.

Table 1 is useful. However, not all entries have clearly attributed sources. Some clinical data require verification. Take the BEVABEL trial, for example. It's listed with an overall response rate (ORR) of 41.2%, but the published interim analysis reported an ORR of 6.0% and a median overall survival of 11.1 months. Each study included in the table should be accompanied by a DOI or PubMed ID for traceability purposes. The review summarises trial results, but doesn't critically assess the quality of the evidence. To make the work stronger, it should distinguish between randomised controlled trials, single-arm phase II studies, and retrospective analyses, while highlighting limitations such as small sample sizes and heterogeneous histologic subgroups. Some clinical metrics, including ORR, ORR-6 and DCR, are reported without consistent definitions, which can cause confusion. It is recommended that the criteria used are clarified (RECIST 1.0 vs 1.1) and that footnotes are included to explain abbreviations in the table. Certain statements appear to generalise treatment efficacy across all ccRCC subtypes, whereas papillary, chromophobe, collecting duct and translocation RCCs have distinct clinical behaviours. Where possible, discussing subtype-specific outcomes would improve accuracy.

Minor issues include repeated content between sections 3.1 and 3.2 and inconsistent terminology, which could be addressed with light editing. The funding statement should specify whether any author has industry relationships or received support related to the drugs discussed, and all abbreviations should be defined at first use. Finally, adding a brief section on the limitations of the review, such as the reliance on early-phase data and heterogeneity of included studies, as well as a short table summarizing key ongoing clinical trials, would further enhance the manuscript.

Reviewer 2 Report

Comments and Suggestions for Authors

Minor Revision Recommendations

While the manuscript is scientifically strong and clearly written, a few minor revisions are suggested to enhance clarity and rigor:

  1. Add a brief methodology section describing the literature search strategy (databases used, timeframe, key inclusion/exclusion criteria). This would improve the transparency expected from a scholarly review.
  2. Eliminate minor redundancies, especially in Sections 3.1 and 3.2, where chemotherapy and targeted therapy are partially repeated.
  3. Refine the discussion of multi-omics subtyping by briefly acknowledging current challenges, such as limited cohort sizes and variability in data integration across studies.
  4. Standardize terminology and formatting (e.g., consistent use of abbreviations like mPFS, ORR, and capitalization of subtype names).
  5. Minor language polishing: a few long sentences could be shortened for better readability and flow.

Reviewer 3 Report

Comments and Suggestions for Authors

Major comments:
1) The authors could develop the limited representation of nccRCC in ongoing immunotherapy trials and how real-world evidence could address this.
2) Please add a paragraph on the emerging role of liquid biopsy or circulating DNA profiling in detecting rare molecularly defined RCCs.
3) In Section 4, the authors should explain how molecular or immune subtypes (e.g., myeloid–lymphoid infiltration, MET-driven tumors) could predict responsiveness to specific agents (TKIs, ICIs, or combinations).
4) The authors could opt to include a summarizing figure connecting major histologic and molecular nccRCC subtypes, their key genomic drivers (MET, FH, TSC1/2, TFEB, SMARCB1, etc.) and preferred therapeutic strategies (targeted, ICI, combination).

Minor comment:
1) Just a few typos: for example, “merely 4% in pRCC” should be “only 4% in pRCC” or “multi-omics” should consistently include a hyphen.

Round 2

Reviewer 3 Report

Comments and Suggestions for Authors

The authors have adequately revised their work taking into consideration the reviewers' suggestions. I am happy to endorse the publication of their revised manuscript.